# Potential Biocontrol Activities of *Populus* Endophytes against Several Plant Pathogens Using Different Inhibitory Mechanisms

**DOI:** 10.3390/pathogens12010013

**Published:** 2022-12-22

**Authors:** Sharon L. Doty, Pierre M. Joubert, Andrea Firrincieli, Andrew W. Sher, Robert Tournay, Carina Kill, Shruti S. Parikh, Patricia Okubara

**Affiliations:** 1School of Environmental and Forest Sciences, College of the Environment, University of Washington, Seattle, WA 98195, USA; 2Department of Plant & Microbial Biology, University of California, Berkeley, CA 94720, USA; 3Department for Innovation in Biological, Agro-Food and Forest Systems, University of Tuscia, 01100 Viterbo, Italy; 4Native Roots School, Taos, NM 87571, USA; 5Department of Food Science and Technology, University of California, Davis, CA 95616, USA; 6Department of Plant Pathology, Washington State University, Pullman, WA 99164, USA

**Keywords:** biocontrol, *Rhizoctonia*, Fusarium, *Pythium*, *Gaemannomyces*, plant pathogens, antimicrobial, endophytes, *Burkholderia*, *Rahnella*

## Abstract

The plant microbiome can be used to bolster plant defense against abiotic and biotic stresses. Some strains of endophytes, the microorganisms within plants, can directly inhibit the growth of plant fungal pathogens. A previously isolated endophyte from wild *Populus* (poplar), WPB of the species *Burkholderia vietnamiensis*, had robust in vitro antifungal activity against pathogen strains that are highly virulent and of concern to Pacific Northwest agriculture: *Rhizoctonia solani* AG-8, *Fusarium culmorum* 70110023, and *Gaemannomyces graminis* var. *tritici* (Ggt) ARS-A1, as well as activity against the oomycete, *Pythium ultimum* 217. A direct screening method was developed for isolation of additional anti-fungal endophytes from wild poplar extracts. By challenging pathogens directly with dilute extracts, eleven isolates were found to be inhibitory to at least two plant pathogen strains and were therefore chosen for further characterization. Genomic analysis was conducted to determine if these endophyte strains harbored genes known to be involved in antimicrobial activities. The newly isolated *Bacillus* strains had gene clusters for production of bacillomycin, fengicyn, and bacillibactin, while the gene cluster for the synthesis of sessilin, viscosin and tolaasin were found in the *Pseudomonas* strains. The biosynthesis gene cluster for occidiofungin (*ocf*) was present in the *Burkholderia vietnamiensis* WPB genome, and an *ocf* deletion mutant lost inhibitory activity against 3 of the 4 pathogens. The new isolates lacked the gene cluster for occidiofungin implying they employ different modes of action. Other symbiotic traits including nitrogen fixation, phosphate solubilization, and the production of auxins and siderophores were investigated. Although it will be necessary to conduct in vivo tests of the candidates with pathogen-infected agricultural crops, the wild poplar tree microbiome may be a rich source of beneficial endophyte strains with potential for biocontrol applications against a variety of pathogens and utilizing varying modes of action.

## 1. Introduction

Pathogenic microorganisms cause crop losses equivalent to billions of dollars annually worldwide. There has been an alarming trend of increasing numbers of new fungal and oomycete plant pathogens [1], with fungal pathogens accounting for nearly 15% of worldwide crop losses each year [2]. The plant pathogens *Rhizoctonia solani*, *Fusarium culmorum*, *Gaeumannomyces graminis* var. *tritici* (Ggt), and the oomycete, *Pythium ultimum*, are widespread pathogens of economically important crops including wheat, rice and barley, grain, legumes, and brassicas worldwide, causing root rot, take-all disease, and damping off [3,4]. Resistant plant varieties take decades to develop and pathogens often develop resistance within several years [5]. Commercial growers rely on a variety of chemicals to control the plant pathogens. However, widespread use of these chemicals comes at high economic and environmental costs such as potentially harming beneficial microorganisms and insects. Additionally, organic farmers are not permitted to use synthetic chemicals. An alternative method is biocontrol that involves the use of non-pathogenic microbial strains to combat the pathogens [5]. Specific strains of bacteria and fungi are known to suppress growth of soilborne fungal pathogens of crop plants under controlled conditions [6,7], but commercialization for the field involves additional criteria [8]. These include targeted (pathogen-specific) activity, durability of activity in a range of soil types and environmental conditions, and application method [9,10]. As such, bio-based products have developed more slowly than synthetic products but represent a growing proportion of the pesticide industry. For soil-borne pathogens, seed treatment is considered to be the application method of choice. Selection of biocontrol microbial isolates or their metabolites for commercialization purposes should ideally be based on multiple pathogen isolates of the same species because genotypes of the plant, pathogen and biocontrol microbe appear to condition the biocontrol interaction [11,12]. Understanding the mode of action of biocontrol metabolites and their impact on the rhizosphere microbiome also will help in the development of a successful bio-based product. The limited number of biocontrol strains in use, however, is largely from the rhizosphere or soil, and as such, could easily be out-competed by other soil microorganisms [13].

Endophytes, the non-pathogenic microorganisms within plants, represent a largely untapped portion of the plant microbiome. In a screen of endophytes isolated from plants growing on the west side of the Washington State Cascade Mountain Range, 21 percent of cultured isolates from poplar (*Populus*) and willow (*Salix*) had in vitro antimicrobial activity against at least 1 pathogen out of 4 tested [14]. This high frequency of anti-microbial activity likely reflects the strong selective pressure in natural systems for resistance to a broad array of pathogens. Because of this pressure towards mutualism with their host, endophytes have developed a number of other traits that increase plant health, growth, yield, and stress tolerance [15,16,17,18]. Bacterial endophytes can systemically colonize plants [19], giving them a key advantage over rhizospheric strains and some fungal endophyte strains which can be highly localized. Endophytes may therefore offer a more stable option for biocontrol compared to existing methods, defending their host from pathogens from within.

Previous studies on potential biocontrol strains from poplar were based on endophytes that had been isolated for other purposes such as nitrogen fixation [14]. For example, several of the *Burkholderia* endophytes exhibited particularly robust inhibition in vitro against several plant pathogens despite the original selection method being growth on nitrogen-limited medium. A particularly robust antimicrobial strain, WPB, described herein, is a strain of *B. vietamiensis*, a member of the *Burkholderia cepacia* complex (BCC), a group found in a variety of ecological niches including soil, plants and animals but has mambers which present health hazards [20]. 

With the hypothesis that direct screening for antifungal activity of poplar endophytes would allow for the isolation of a wider variety of potential biocontrol strains, a novel method for screening plant extracts for antifungal endophytes was developed, without any pre-screening or isolation steps required. Using this method, we isolated a variety of new endophyte strains from wild poplar with in vitro antifungal activities and report the genomic analysis on the potential inhibitory mechanisms. Genomic analysis of *B. vietnamiensis* strain WPB indicated that occidiofungin, a known antimicrobial compound, could be key to its effectiveness. Using targeted mutagenesis and phenotypical and biochemical confirmation, it was shown that WPB inhibits pathogens through multiple modes of action. Here, we also report on the genomic analysis of additional endophytes isolated from poplar that inhibited selected pathogen strains in vitro. The goal of this study was to isolate strains that employ different mechanisms of antimicrobial activities, reducing the risk of pathogens developing resistance. 

## 2. Results

### 2.1. Antimicrobial Activities of Strain WPB

*Burkholderia vietnamiensis* strain WPB was originally isolated in a screen for nitrogen-fixing endophytes of wild poplar [21]. The strain produces auxin and promotes plant growth under nitrogen-limited conditions [22]. To determine if the strain also had potential biocontrol activity against plant pathogens, the dual plate inhibition assay was used against four virulent strains of agricultural pathogens of concern in the Pacific Northwest. In vitro assays indicated that WPB strongly inhibited the growth of *Rhizoctonia solani* AG-8, *Fusarium culmorum* 70110023, and Ggt ARS-A1 while the oomycete *Pythium ultimum* 217 was inhibited by WPB minimally (Figure 1, Table 1).

### 2.2. Isolation and Screening of New Anti-Fungal Endophytes

Strain WPB and our other previously isolated endophyte strains with in vitro antifungal activities [14] had been selected based on other plant symbiotic traits. To more directly isolate strains with potential biocontrol activity, hundreds of endophytes were isolated from wild poplar trees and screened specifically for in vitro antifungal activity against *R.solani* AG-8. The active strains were then tested against the other plant pathogens using the dual plate assay. Measurements of the zone of inhibition were recorded and then converted into a qualitative scale for ease of comparison between the different fungal assays (Table 2). All of the endophytes had some level of activity against at least two plant pathogen strains, but the strength of the inhibitory response varied from pathogen to pathogen. For example, strain AFE8 demonstrated a strong inhibitory effect on *R. solani* AG-8 but a weaker effect on the growth of *F. culmorum* 70110023 or Ggt ARS-A1. The oomycete, *P. ultimum* 217, appeared to be more resistant to endophyte-induced inhibition, with none of the strains being capable of strong inhibition. Ggt, on the other hand, was strongly inhibited by 7 of the 11 endophytes. Only two endophytes, AFE4A and AFE5 inhibited all four of the pathogen strains tested. The eleven most active endophyte strains were selected for strain identification by rDNA sequencing (Table 2). 

### 2.3. Analyses for Potential Antimicrobial Mechanisms 

#### 2.3.1. In silico Analysis of Antimicrobial Biosynthetic Gene Clusters in Strain WPB

The antiSMASH analysis predicted several biosynthetic gene clusters (BGCs) with antimicrobial activities in WPB, including clusters for occidiofungin A, the polyketide cepacin A, and the siderophore ornibactin (Table 3). The antiSMASH analysis did not identify any BGCs for other antifungal compounds, such as pyrrolnitrin, cepacidine A, and the putative lipopeptide AFC-BC11, that have been identified in other members of the BCC [23]. Occidiofungin, originally isolated from *Burkholderia contaminans* MS14, has broad-spectrum antifungal activity against plant and animal pathogens [24]. WPB contains an occidiofungin A cluster with 94% similarity to the MIBiG reference gene cluster from *Burkholderia pyrrocinia* Lyc2 (Figure 2), a plant-associated strain showing strong antifungal activity against *Aspergillus flavus, Cladosporium sp, Cochliobolus heterostrophus, Colletotrichum acutatum, Gaeumannomyces graminis var. tritici, Geotrichum candidum, Glomerella cingulata* and *Thielaviopsis basicola* [25]. 

WPB also contains a BGC for the siderophore ornibactin, which has been shown to have antibacterial activities against the plant-pathogenic bacteria *Erwinia amylovora*, *Xanthomonas citri* pv. malvacearum, and *Clavibacter michiganensis* subsp. *michiganensis* [26]. In members of the BCC, ornibactin has been implicated in complications in immunocompromised individuals, particularly those with cystic fibrosis [27]. While the ornibactin cluster in WPB has a 93% genes similarity to the *B. cepacia* 89 reference cluster (Figure 2), it lacks the *orb*K gene, which is consistent other members of the *B. vietnamiensis* species, including the type strain *B. vietnamiensis* LMG 10929, a plant-associated strain isolated from the rhizosphere of rice [28].

The third BGC identified in WPB, the polyketide cepacin A, a hserlactone with potent activity against both Gram-positive bacteria and the plant pathogen *Pythium ultimum*, the parasitic oomycete formerly classified as fungi [23]. While the cepacin A BGC in WPB shares only a 62% gene similarity with the *B. ambifaria* IOP40-10 cepacin A reference cluster in MIBiG (Figure 2), it shares high similarity with other *B. vietnamiensis* strains, including the type strain LMG 10929 (Appendix A
Figure A1). The *B. vietnamiensis* cepacin A clusters contained two hypothetical proteins of unknown function, as well as several biosynthetic, transport, and regulatory genes not found in the *B. ambifaria* cluster (data not shown).

#### 2.3.2. In Silico Analysis of Antimicrobial Biosynthetic Gene Clusters in the AFE Strains

The genomes of the AFE strains were mined using antiSMASH to detect BGCs putatively involved in the antifungal activity observed in the plate inhibition assays. AFE strains showing low or no antifungal activity were also included in this analysis since BGC can be silent under the experimental condition used for testing [29]. As expected, the AFE strains with the highest number of BGCs were those belonging to the species *Pseudomonas* and *Bacillus*, while *Pseudomonas* AFE16 and the *Enterobacteriaceae* AFE strains carried the lowest number of BGCs. All the *Pseudomonas* strains, except AFE16, carried at least one BGC involved in the synthesis of a secondary metabolite with antimicrobial activity (Table 3). Specifically, *Pseudomonas* sp. AFE8 carried one BGC sharing 100% similarity score against the viscosin BGC from *Pseudomonas fluorescens* SBW25, and a BGC showing a 60% similarity with the tolaasin I / tolaasin F from *Pseudomonas costantinii* DSM 16734 (Figure 3). When compared to the tolaasin reference cluster, the BGC from AFE8 shows only a partial conservation of core and auxiliary biosynthetic genes, suggesting that this cluster could be involved in the synthesis of a tolaasin-like non ribosomal peptide with a different structure and activity from tolaasin (Figure 3). AFE5 carried two BGCs, respectively involved in the synthesis of tolaasin and sessilin, and sharing a score similarity of 80 and 66%. These two clusters were both detected at the edges of contigs. A close inspection indicates that sessilin and tolaasin BGC in AFE5 are part of a single cluster involved in the synthesis of sessilin (Figure 3). Tolaasin, sessilin, and viscosin are well-characterized non-ribosomal peptides with antifungal activity. The tolaasin was first characterized in the fungal pathogen *Pseudomonas tolaasii* which is the causative agent of the Brown blotch disease, while viscosin and sessilin have been found to be effective against *Aspergillus fumigatus*, *Batrachochytrium dendrobatidis, Rhizoctonia solani,* and *Pythium myriotylum* [30,31].

The *Bacillus* AFE strains AFE4A and AFE21B carried the highest number of BGCs with antifungal and antimicrobial activity. Both AFE4A and AFE21B shared one copy of the BGCs difficidin, macrolactin H, bacillaene, fengycin, bacilysin, bacillomycin D and surfactin. Among these, macrolactins, bacillomycin and fengycin have been shown to inhibit the growth of several plant pathogens including *Fusarium graminearum*, *Botrytis cinerea, Colletotrichum acutatum, Rhizoctonia solani, Magnaporthe* grisea, and *Gaeumannomyces graminis* var. *tritici* [32,33,34]. Interestingly, in both AFE4A and AFE21B, the bacillomycin D and fengycin BGCs were located in the same chromosomal region, suggesting that these antifungal agents could act synergistically in the inhibition of the fungal pathogen (Figure 4).

#### 2.3.3. Occidiofungin Mutant of WPB

Gu and colleagues demonstrated through Tn5 mutagenesis of *Burkholderia contaminans* that an occidiofungin gene cluster was required for antifungal activity against *Rhizoctonia solani*, *Pythium ultimum* and others [35]. Since strain WPB contains the same gene cluster, we chose one of the genes, *ocf*E, as a mutagenesis target to determine if the compound was responsible for the in vitro antimicrobial activities we observed. Extracts from the wild-type and mutant strains were tested by mass spectrometry. Wild-type WPB grown with *R. solani* AG-8 produced a higher level of occidiofungin than when grown without the fungus, indicating it may be inducible (Figure 5, comparing black and blue chromatograms). Furthermore, the mass spectrometry results confirmed that the *ocf* mutant did not produce occidiofungin (Figure 5, red chromatogram). The *ocf* mutant retained minimal ability to inhibit *R. solani* AG-8 growth (Table 1 and Figure 1), implicating this pathway as a major mechanism this strain uses to directly block pathogenic fungi. This was further supported by the loss of inhibition by the mutant toward the fungi *F. culmorum* 70110023 or *P. ultimum* 217. However, the mutant retained inhibitory activity against Ggt ARS-A1, indicating that a different mode of action is utilized (Figure 1).

#### 2.3.4. Confirmation of the Lack of Occidiofungin from AFE Strains by LC-MS-MS

Since occidiofungin was a key antimicrobial compound from strain WPB, the AFE strains were also screened by mass spectrometry to confirm in silico gene analysis indicating a lack of the occidiofungin gene cluster. None of the strains displayed a peak indicative of occidiofungin (data not shown).

### 2.4. Symbiotic Traits of the AFE Strains

Siderophore production is a common trait of plant-associated microorganisms that is thought to play a biocontrol role by sequestering iron away from pathogens and preventing their growth. The siderophore production assay was used to determine this activity in the endophytes. Seven of the eleven strains exhibited strong to very strong siderophore production (Table 4).

Tests indicative of other symbiotic traits including phytohormone production, nitrogen fixation and phosphate solubilization were performed on the 11 active strains. Strains AFE 3, 9, 14, and 22 were strong producers of auxins (Appendix A
Figure A2) so they may enhance root growth. Most of the strains were able to solubilize tricalcium phosphate, especially strains AFE 1, 3, and 14. Though solubilization of other phosphate salts such as iron phosphate and aluminum phosphate should be tested [36], solubility of tricalcium phosphate can be an indicator of the ability to increase the bioavailability of phosphate, an important macronutrient [37]. Strains AFE3 and 9 produced a nitrogenase gene PCR product and were positive in the acetylene reduction assay for nitrogenase activity (Appendix A
Figure A3).

### 2.5. Phylogenomic Analysis of WPB and AFE Strains for Suitability for Use in Agriculture

For the strains to be used directly in agriculture, they must not be phylogenetically affiliated to known human and plant pathogens. Therefore, a phylogenomic analysis was carried out to place WPB and the AFE strain into clusters of phylogenetically related genomes at the species level. 

Whole-genome taxonomic classification identified WPB as *B. vietnamiensis*, matching most closely with the type-strain *B. vietnamiensis* LMB 10929, and clustering with other members of the BCC. Additionally, WPB was predicted by PathFinder as being a potential human pathogen with high probability (0.843), identifying 246 sequences belonging to a pathogenic protein families and 24 sequences belonging to non-pathogenic protein families.

All Enterobacteriaceae AFE strains were included in already known species clusters (Table A2). The *Rahnella* strains AFE22 and AFE3 were identified as *R. variigena* and *R. victoriana*, respectively, and the reference strains of the species are potential plant pathogens being isolated from various oak species exhibiting symptoms of Acute Oak Decline (AOD) and exhibited a mild hypersensitive response on tobacco plants [38]. On the other hand, the AFE9 strain was identified as *Rahnella aceris*, which includes only non-pathogenic plant-associated *Rahnella* isolated from *Acer pictum* sap [39]. The last *Enterobacteriaceae* strain, *Erwinia* AFE1, was identified as *Erwinia billingiae*. While *Erwinia* species include several renowned plant and human pathogens, the representative strain of the *E. billingiae* species cluster lacks several virulence-associated genes found in the plant pathogens *E. amylovora*, *E. pyrifoliae*, and *E. piriflorinigrans* [40,41]. The *E. billingiae* species clusters include 7 genomes from soil and host-associated (plants and *Caenorhabditis elegans*) isolates. 

The *Bacillus* strains AFE21B and AFE4A were identified as members of the *Bacillus velezensis* species (Table A2) which include strains already proposed to EPA and EFSA for use in agriculture as a biopesticide, and as an additive in animal feed [42,43].

*Pseudomonas* strains AFE8 and AFE5 were affiliated to species-clusters of the *P. fluorescens* complex which currently includes 48 known *Pseudomonas* species. Specifically, AFE8 and AFE5 were classified as *P. lurida* and *P. kitaguniensis*, respectively (Table A2). The *P. lurida* species-cluster consists of 25 *Pseudomonas* strains mostly identified as environmental and host-associated (*C. elegans*) isolates, while only one strain, i.e., AU10973 (GCA_000801835.1), was isolated from cystic fibrosis (CF) patients [44]. While *P. fluorescens* strains are not considered an etiologic agent of pulmonary diseases, members of the *P. fluorescens* species complex are identified at a low frequency in clinical samples from CF patients. It is worth mentioning that given their extreme metabolic versatility, members of the *P. fluorescens* complex are naturally occurring at low levels in the indigenous microbiota of various body sites, and are also moderately abundant in the respiratory microbiota [45]. Despite this, *P. fluorescens* is rarely associated with acute infections [46]. Finally, the *Pseudomonas* strain AFE5 was placed within a species cluster representative of 5 plant-associated *Pseudomonas*, with the isolate MAFF 212408 being type-strain for the species *P. kitaguniensis*. The *P. kitaguniensis* strain MAFF 212408 (GCF_009296165.1) was identified as the causative agent of the ‘bacterial rot disease of Welsh onion’.

## 3. Discussion

The goal of this study was to isolate endophyte strains that could inhibit key pathogen strains of concern in the Pacific Northwest. To select for such activities, it is necessary to conduct the initial testing in vitro. Once candidate strains are identified, the next phase is to complete genomic sequencing and analysis for genes known to be involved in human pathogenicity since the ultimate goal is to use the inhibitory strains in agriculture. A case in point is that, in a previous study, the endophytes showing the strongest in vitro antifungal activity were of the genus *Burkholderia* [14]. *Burkholderia* have been studied for their abilities in plant growth promotion, phytoremediation, endophytic nitrogen fixation, reducing plant abiotic stresses and as biocontrol agents [47,48,49,50,51], while others are known as human pathogens [20]. Since the original *Burkholderia* genus includes such diverse species, molecular signature and phylogenomic analysis were performed that divided them into *Burkholderia* and *Paraburkholderia* genera [52]. A number of root-associated *Burkholderia* strains also lacked genes associated with human pathogenicity [53]. Subsequently, a new genus, *Caballeronia* gen. nov.was proposed that contains non-pathogenic and plant-associated members of both *Burkholderia* and *Paraburkholderia* [54]. The previously isolated strain of *Burkholderia vietnamiensis*, WPB, was included in this study of potential mechanisms of antimicrobial activities, although it would not be used in agriculture since this strain falls within the *Burkholderia* BCC and was determined to harbor genes associated with human pathogenicity. Nonetheless, it served to provide valuable insight into inhibitory mechanisms.

Interestingly, when studying poplar leaves growing along the Snoqualmie River in Washington State, nearly one-fourth of the bacterial microbiome was found to be of the *Burkholderia/Paraburkholderia* genus (unpublished data). In a recent study, the *Burkholderiaceae,* along with *Pseudomonadaceae* and *Enterobacteriaceae*, was one of the most abundant families detected in the core microbiome of wild poplar inhabiting xeric riparian zones, and riparian zones with mid hot-dry and moist (mesic) climates [55]. Our study further supports the concept that antimicrobial/antifungal producing strains are a core component of the poplar microbiome, and not linked to a specific environment. The high prevalence of *Burkholderia* points to the importance of this genus in the plant microbiome. 

In an effort to isolate a broader biodiversity of endophytes, a different technique was employed in this study. Rather than testing individual isolates for antifungal activity, poplar extracts containing endophytic microorganisms were challenged directly against a fungal pathogen. Microbes were then isolated from zones of inhibition. Though all the strains were originally selected for the ability to inhibit *R. solani* AG-8, many of them also inhibited other fungal pathogen strains regardless of evolutionary distance. Due to the vast evolutionary distance from the fungi to oomycetes, it was not surprising that few of the strains had activity against *P. ultimum*. However, this method of challenging extracts rather than single isolates could be applied to any pathogen strains of interest to rapidly select potential biocontrol strains. Though strains AFE 3, 14, and 22 were originally isolated with apparent inhibitory activity against *R. solani AG-8*, when fully purified, they were no longer active against this pathogen. It is possible that they required a microbial partner or phytochemicals mimicking host-like conditions that were lost during the strain purification phase. For instance, the expression of T6SS, which can be used to deliver antifungal effectors, is silent unless host-like conditions are mimicked [56]. 

One goal of this study was to isolate strains that each employed different mechanisms of inhibiting plant pathogens. Application of a consortium of strains with complementary antimicrobial activities would be ideal in biocontrol situations, allowing for possible synergies between modes of action and reducing the risk of pathogens developing resistance. In this respect, it has been demonstrated that a combination of three biocontrol species was superior for inhibiting *Fusarium* [57]. Some of the antimicrobials produced by the *Pseudomonas* AFE strain have similar modes of action. For instance, the most likely action of the cyclic lipodepsipeptides (CLPs) of the Viscosin-groups involves the direct interaction with the cellular membrane through pore-formation [58]. A similar activity was observed for the lipopeptides of the Tolaasin-group which have the ability to form ion channels into lipid bilayer [59]. Despite this, prior to the development of a microbial consortium with enhanced antifungal potential, the positive interaction between members of the consortium must be assessed. Indeed, *Bacillus* and *Pseudomonas* could compete for the same niche, where *Bacillus* lipopeptides are used to counteract and reduce the toxicity of the lipopeptides synthesized by *Pseudomonas* [60].

In this study, AFE4A was the best performing strain in terms of breadth and strength of in vitro antifungal activities. AFE4A typically had wide zones of inhibition implying an antifungal mechanism based on the secretion of multiple secondary metabolites with antifungal activity. This is in partial agreement with the antiSMASH analysis which identifies AFE4A, and also AFE21B, as the AFE strains with the greatest antifungal potential. AFE4A and AFE21B were isolated from wild poplars inhabiting two different river systems. Despite this, both strains carried the same BGC’s with antifungal activities and yet had strikingly different inhibition patterns. This would provide an opportunity to study the molecular mechanisms underlying the biosynthesis of the antifungals rather than focusing on comparative genome analysis. The presence of a single lanthipeptide antibiotic gene cluster was found to distinguish AFE4A from AFE21B. Since AFE4A had the most widespread activity against all pathogens tested and the most effective antifungal activity, this defining lanthipeptide (Lantibiotics) gene cluster merits further investigation. However, all the Lantibiotics characterized so far in *Bacillus* species possess antimicrobial activity, while Pinesin A and B, which are produced by the Gram-negative *Chitinophaga pinensis,* are the only known Lantibiotics with antifungal activity [61,62]. Therefore, the differences observed between AFE4A and AFE21B could be explained only by different antifungal mechanisms or by differing levels of expression of the BGC’s.

The genome of the yeast strain AFE11, *Aureobasidium pullulans*, was not sequenced. However, *A. pullulans* is known to produce a variety of antimicrobial compounds including toxins, volatile compounds, degradative enzymes, and siderophores [42,63]. Described as being a generalist biocontrol strain, it may be the simultaneous production of such a diverse arsenal that leads to its success.

The genome of WPB was found to contain the biosynthetic genes required for the production of occidiofungin, a known antifungal compound. Mutagenesis of the biosynthetic gene *ofc*E in WPB highlighted the importance of this antimicrobial to the powerful inhibition exhibited by this strain. The results also indicated that other inhibitory activities are at play, although the mechanism still needs to be resolved as no additional BCGs with known antifungal activities were found in WPB. As a genus, *Burkholderia* possess large, complex, and variable genomes and ongoing efforts to identify novel antimicrobial compounds will continue to benefit from rapidly growing libraries of sequenced genomes, including the sequencing and assembly of the genomes from 450 members of *Burkholderiaceae* completed by Mullins et al. (2020) [64]. The *ofc* mutant retained strong inhibition of Ggt ARS-A1 despite the lack of occidiofungin production. Genomic analyses and confirmation through LC-MS has shown that the AFE strains also lack the capability to produce occidiofungin. This implies antifungal mechanisms in all strains tested that are distinct from occidiofungin. 

It may be that endophytes are uniquely positioned to evolve antifungal mechanisms that are targeted specifically against pathogens. *Populus* associates with both ectomycorrhizal and arbuscular mycorrhizal fungi; therefore, forcing co-evolution of the bacterial endophytes with these essential fungal partners. Bacterial endophytes must therefore be capable of co-habitation with these beneficial fungi despite the anti-fungal activities demonstrated here against pathogens. Living with a host plant would apply a heavy evolutionary pressure to develop mechanisms to differentiate between pathogens and beneficial endophytic fungi. The unique niche occupied by endophytes has encouraged the development of several other beneficial traits including nitrogen fixation, phosphate solubilization, and phytohormone production. Though the strains isolated in this study were selected for the ability to inhibit pathogens, several had these beneficial traits as well. Endophytes are a rich resource with great potential to improve the resilience of crops to environmental challenges, both abiotic and biotic.

## 4. Materials and Methods

### 4.1. Pathogen Strains

The pathogen strains were sourced and maintained as described [14]. *Rhizoctonia solani* AG-8 isolate C1 and *Fusarium culmorum* 70110023 were maintained on potato dextrose agar (PDA). *Pythium ultimum* 217 was cultured on SY agar. *Gaemannomyces graminis* var. *tritici* ARS-A1was maintained on 1/5 X PDA. The pathogen strains used in this study are highly virulent and of concern to Pacific Northwest agriculture.

### 4.2. Endophyte Extraction from Plant Tissue

Anti-fungal endophyte (AFE) strains were isolated from wild poplar stems collected from the riparian zones of the Snoqualmie, Skykomish, and Yakima Rivers of Washington State, USA. Branches were surface-sterilized with 10% commercial bleach for 10 min and 1% Iodophor for 5 min, followed by three sterile water rinses. Tissue was ground directly into rich media using a mortar and pestle. Extracts were transferred to sterile conical tubes and large particles allowed to settle out before use. 

### 4.3. High-Throughput Antifungal Screen and Inhibition Assays

Endophytes from the poplar extracts were screened for activity before isolation by pipetting diluted plant extract in a circle around the entire perimeter of PDA plates, which had been centrally inoculated with *Rhizoctonia solani* AG-8. As the pathogen grew, reaching the extract along the border, zones of inhibition were observed. The mixed population bacterial growth central to the inhibition was then streak-purified on MGL agar with incubations at 30 °C. Purified strains were tested for antifungal activity using the dual plate inhibition assay as described [14]. New isolates with antifungal activity were cryogenically stored in glycerol at −80 °C and named with the AFE designation. All subsequent testing of inhibition by the AFE strains as well as WPB and the WPB *ocf* mutant were done through the dual plate inhibition assay as well. Individual assays were performed in duplicate (two bacterial spots per plate) with three plates per assay. The Petri dishes were 100 × 15 mm.

### 4.4. Endophyte Identification

Strain WPB was previously isolated from wild poplar [21] and described [22]. New AFE strains were identified through PCR amplification of the 16S rRNA gene of each strain, or 28S rRNA gene of AFE11, using 8F (AGAGTTTGATCCTGGCTCAG′) and 1492R (GGTTACCTTGTTACGACTT-3) primers, or D1/D2 (F63 5′-GCATATCAATAAGCGGAGGAAAAG-3′ and LR3 CGTCCGTGTTTCAAGACGG) for the 28S gene of AFE11. The genetic sequences were then compared to those in the NCBI’s GenBank using the BLASTN webserver.

### 4.5. Genomic Sequencing and Analysis

The WPB genome had previously been sequenced and assembled at the Department of Energy Joint Genome Institute (JGI) as part of the research topic “Defining the poplar root microbiome”. The assembled genome was submitted to the Rapid Annotation Subsystems Technology (RAST) [65] for annotation and the predicted protein-encoding genes (PEGs) submitted to KEGG KofamKOALA [66] for functional annotation and KEGG Ortholog (KO) assignment. Genome annotation, performed at the JGI, was carried out using the DOE-JGI Microbial Annotation Pipeline (DOE-JGI MAP) [67]. Whole-genome-based taxonomic classification was completed via the Type (Strain) Genome Server (TYGS) [68], and the potential for human pathogenicity predicted using PathogenFinder (v1.1) [69] at the Center for Genomic Epidemiology. The WPB genome is available through the NCBI GenBank database under accession GCA_900102765.1

Of the new AFE isolates, only endophytes showing strong in vitro inhibitory effects against at least one fungal crop pathogen were selected for genomic sequencing, species identification, and in silico analysis of antifungal gene clusters. Strains AFE 1, 3, 4A, 5, 8, 9, 16, 21B and 22 were grown in 10mL MG/L broth, pelleted, and resuspended in Tris buffer (pH 8). Cells were lysed via incubation with 10% SDS (sodium dodecyl sulfate) and 5M sodium chloride at 68C for 30 min. Following RNaseA and Proteinase K treatments, the samples were subjected to phenol-chloroform extractions and ethanol precipitation. The paired-end libraries were constructed from approximately 376 ng of gDNA using the Nextera DNA Flex Library preparation kit, and loaded in one flow cell. Each library was barcode sequenced using a 2 × 250-bp format. The MiSeq run was performed using the MiSeq Reagent Kit v3 (600 cycles) chemistry. The following steps were then performed for assembly: (1) BBMap package was used to remove adapters and keep high-quality reads; (2) filtered reads were assembled using SPAdes (version v3.13.0; ––phred–offset 33 ––cov–cutoff auto –t 16 –m 64 ––careful –k 55,97,127) [70]; (3) contigs were discarded if the length was <1kbp (BBTools reformat.sh: minlength).

Assembly and annotation stats are reported in Appendix A
Table A1. Open reading frame (ORF) prediction was performed using RAST [71]. For the taxonomic classification of the AFE strains we used the Genome Taxonomy Database (GTDB-Tk) version 2.1.0 [72].

### 4.6. In Silico Analysis of Antimicrobial Biosynthetic Gene Clusters in WPB and AFE Strains 

The presence of biosynthetic gene clusters (BGCs) encoding for putative non-ribosomal peptidase synthetase (NRPS), polyketide synthases (PKSs), and other enzymes involved in the synthesis of secondary metabolites with predicted antimicrobial activities was performed by using the antiSMASH (v6.1.1) [73]. The analysis included the antiSMASH options for MIBiG comparison, ClusterBlast and KnownClusterBlast similar gene clusters, and the Pfam and TIGRFAM domains. 

### 4.7. Occidiofungin Mutant Construction

A plasmid was first constructed for directed mutagenesis of endophyte strains. To ensure the selectable marker would be functional in the endophyte, the kanamycin resistance gene was cloned from pBHR-GFP into the EcoRI site of pUC18 using pGEM-T-Easy (Promega), and named pMUT0. Using the approach of Zuniga and colleagues [74], the *ofc*E gene of WPB was targeted. Primers were designed to amplify a 1Kb portion of the *ocf*E gene from WPB, and *Bam*HI sites were added to the primers to facilitate cloning into the *Bam*HI site of pMUT0. Sequences of the primers were as follows: 

Forward primer, TAACGAAAGCTTAAACCGTGTACGACGCCTAT

Reverse primer TAACGAGGATCCCTGCTGAAGTACGCATCCGA 

The resulting plasmid was purified and electroporated into electrocompetent WPB cells. Kanamycin resistant colonies were verified by PCR to have a disruption of the *ofc*E gene.

### 4.8. Chemical Extraction and Analysis of Secreted Antimicrobials 

*Rhizoctonia solani* (RS) was inhibited by strain WPB and most of the AFE bacterial strains (AFE 1, 3, 4A, 5, 8, 9, 16, 21B, and 22), and was therefore chosen for antifungal chemical discovery. *R. solani* was grown on PDA (potato dextrose agar). Square plugs (0.8 cm) were then taken from the active margin of *R. solani*, and transferred to new PDA plates. After 24 h of fungal growth, 4 drops (2.5 microliters per drop) of a 0.2 OD_600_ culture were added equidistantly to the periphery of the plate. Plates were incubated for another 6 days at 20 °C before the zone of inhibition was cut out for extraction. In order to extract the most compounds, a mixture of acetonitrile, water, and methanol (ACN:H2O:MeOH; 2:1.5:0.5) was used with 0.1% formic acid. Extracts were run on the AB Sciex 5600 QTOF Tandem Quadrupole mass spectrometer in positive mode using reverse phase chromatography at the University of Washington Medicinal Chemistry Mass Spectrometry Center. The occidiofungin peak was determined by the known masses of occidofungin A and B [24].

### 4.9. Assays for Symbiotic Traits 

The production of auxins as well as tricalcium phosphate solubilization were quantified as described [14]. The presence of the nitrogenase subunit *nif*H gene was determined as described [14]. Nitrogen fixation was assessed using the acetylene reduction assay as follows. Cell suspensions were adjusted to an optical density (OD_600_) of 0.4 in nitrogen-free medium. One hundred microliters were transferred to 15mL amber vials containing 6ml of NL-CCM agar [75]. Vials were dosed with 0.1ml acetylene gas and incubated for 3 days at 30C. Headspace was analyzed with a gas chromatograph equipped with a flame ionization detector as described [76]. Strain WP5 [21], a diazotrophic endophyte, was included as a positive control. AFE3 and AFE9 were tested in triplicate while the remaining AFE strains that were all negative by PCR for the presence of the nitrogenase gene were tested in singlet and in total served as negative controls.

## 5. Conclusions

This study expands our understanding of the potential for bacterial endophytes to be used in biocontrol applications. Direct in vitro screening for antimicrobial activity led to the discovery of many endophytes of poplar that had potential for use in biocontrol. While occidiofungin was determined to be the primary mode of action for the *Burkholderia* strain, the lack of this biosynthetic gene cluster in the new isolates points to different mechanisms at work. Several of the strains harbored genes encoding known antimicrobial compounds, while others had none, indicating novel antifungal pathways yet to be discovered. These varying and perhaps novel mechanisms of antimicrobial activities may be required to reduce the risk of the development of resistance by pathogens. The next phase of this research would be to conduct in vivo tests with pathogen-infected agricultural crops to determine if consortia of strains with these in vitro pathogen inhibition activities will protect the host plants. It will also be necessary to test more strains of each pathogen species to assess the applicability. In addition to their biocontrol potential, some of the endophyte strains could have the ability to be beneficial in other ways such as providing essential nutrients, including nitrogen and phosphorus. With the ability of many endophytes to colonize a broad spectrum of crop species, and to colonize systemically, the endophytes studied here likely have the tools required to chemically defend their host while also strengthening it from within.

## 6. Patents

The work was not patented but reported in the University of Washington Report of Innovation #48629.

## Figures and Tables

**Figure 1 pathogens-12-00013-f001:**
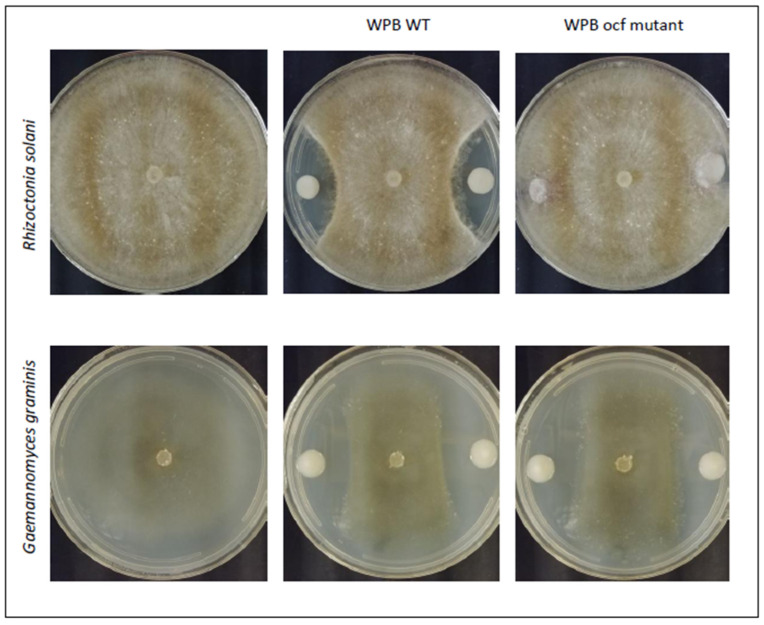
**Example in vitro antifungal activities**. Wild-type (WT) *Burkholderia vietnamiensis* strain WPB and the *ocf* mutant against *Rhizoctonia solani* AG-8 and *Gaemannomyces graminis* var. *tritici* ARS-A1.

**Figure 2 pathogens-12-00013-f002:**
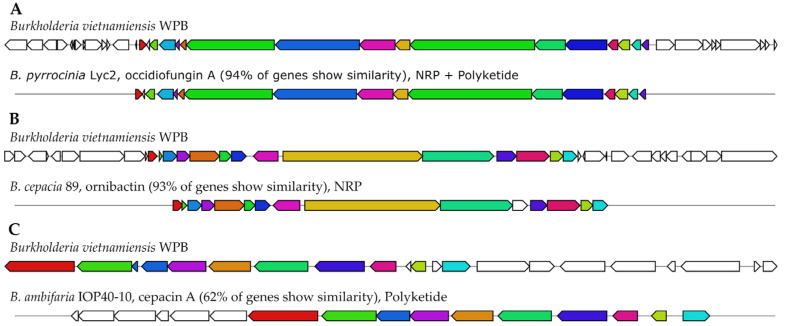
**Organization of the biosynthetic gene clusters identified in *Burkholderia vietnamiensis* WPB.** Orthologous genes shared between the BGCs in WPB and reference strain are shown with the same color for (**A**) occidiofungin (**A**,**B**) ornibactin, and (**C**) cepacin A. The genes not shared between WPB and the reference strain are in white.

**Figure 3 pathogens-12-00013-f003:**
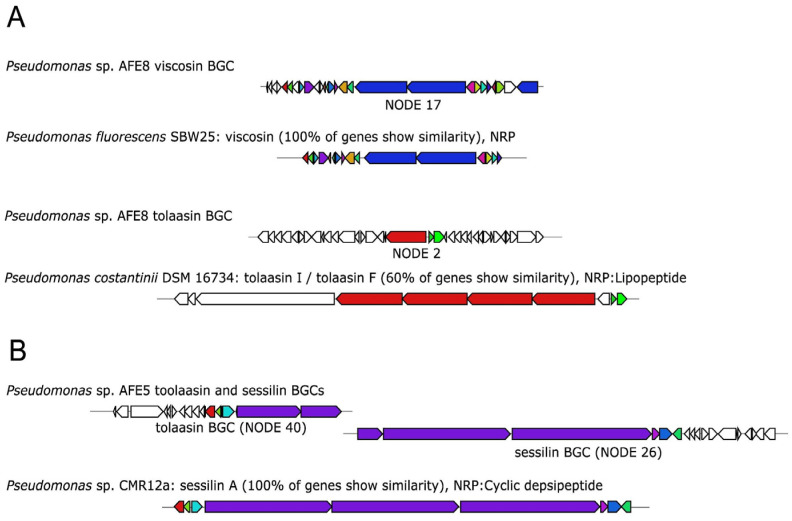
**Organization of the biosynthetic gene clusters identified in the *Pseudomonas* strains AFE8** (**A**) **and AFE5** (**B**). Orthologous genes shared between the BGCs in the AFE and reference strain are shown with the same color, while the genes not shared are in white.

**Figure 4 pathogens-12-00013-f004:**
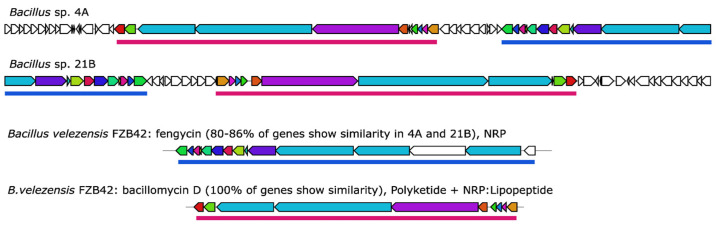
**Topology of the fengycin and bacillomycin D gene clusters in the *Bacillus* AFE strains 4A and 21B.** The blue and dark-pink lines indicate the fengycin and bacillomycin D gene clusters, respectively. Orthologous genes shared between the BGCs in the AFE and reference strain are shown with the same color, while the genes not shared are in white.

**Figure 5 pathogens-12-00013-f005:**
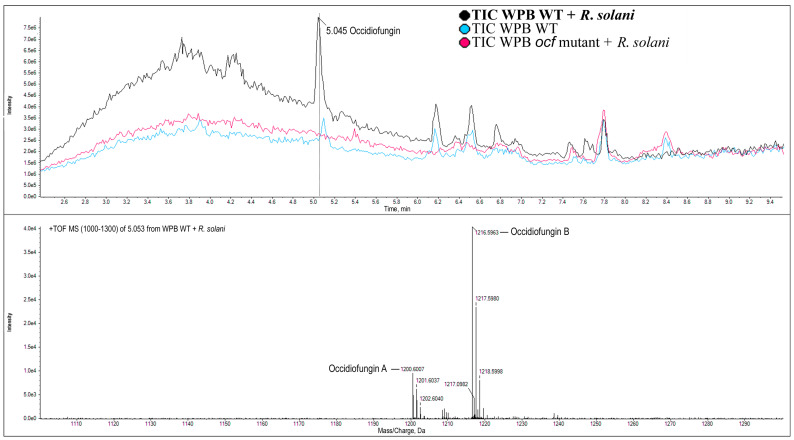
**Confirmation of occidiofungin production in strain WPB by mass spectrometry.** Top panel: Black line indicates wild-type WPB grown with *Rhizoctonia solani* AG-8; red line indicates the *ocf* mutant; blue line indicates wild-type WPB grown in the absence of fungus. Vertical line indicates the occidiofungin peak. Lower panel: M+H molecules present in the occidiofungin peak.

**Table 1 pathogens-12-00013-t001:** **Zones of inhibition from wild-type *Burkholderia vietnamiensis* strain WPB and the occidiofungin *ocf* mutant.** The distances from the bacterial colony and the pathogen strains were quantified. Units are in millimeters. Standard deviation is indicated by ±.

Strain	*R. solani*	Ggt	*F. culmorum*	*P. ultimum*
WPB	5.03 ± 0.64	11.40 ± 1.25	4.44 ± 0.87	2.17 ± 0.98
WPB ocf	0.00 ± 0.00	5.46 ± 0.61	0.00 ± 0.00	0.00 ± 0.00

**Table 2 pathogens-12-00013-t002:** **Antifungal Endophytes.** Species identification, sources and in vitro antifungal activity of the eleven chosen Anti-Fungal Endophyte (AFE) strains as determined by the dual plate assay [-, no activity; + very weak inhibition; ++ weak; +++ strong; ++++ very strong]. The pathogen strains were *R. solani* AG-8, Ggt ARS-A1, *F. culmorum* 70110023, and *P. ultimum* 217.

Strain	Species	Source	*R. solani*	Ggt	*F. culmorum*	*P. ultimum*
AFE 1	*Rahnella* sp.	Snoqualmie River wild poplar 4	++	++++	++	−
AFE 3	*Rahnella* sp.	Snoqualmie River wild poplar 4	−	++++	−	+
AFE 4A	*Bacillus velezensis*	Snoqualmie River wild poplar 4	++++	++++	+++	+
AFE 5	*Pseudomonas* sp.	Snoqualmie River wild poplar 4	+++	++	+	+
AFE 8	*Pseudomonas* sp.	Yakima River wild poplar 12	+++	++	++	−
AFE 9	*Rahnella aquatilis*	Yakima River wild poplar 12	++	+++	−	++
AFE 11	*Aureobasidium pullulans*	Skykomish River wild poplar	+	++	−	−
AFE 14	*Pantoea agglomerans*	Skykomish River wild poplar	−	++	−	−
AFE 16	*Pseudomonas graminis*	Skykomish River wild poplar	+	+++	−	−
AFE 21B	*Bacillus velezensis*	Skykomish River wild poplar	++	++++	++	−
AFE 22	*Rahnella* sp.	Skykomish River wild poplar	−	+++	−	+

**Table 3 pathogens-12-00013-t003:** **Known antimicrobial biosynthetic gene clusters.** WPB and the AFE strains carry a variety of biosynthetic gene clusters with known antifungal and antibacterial activity.

Strain	Total # of BGC	BGC’s with Known Antimicrobial Activity ^1^	Activity
WPB	12	cepacin A (62%)	antibiotic/antioomycetes
		ornibactin (93%)	antibiotic
		occidiofungin (94%)	antifungal
AFE 1	7	-	
AFE 3	6	-	
AFE 4A	21	difficidin ^2^ (100%)	antibacterial
		macrolactin (100%)	antibacterial/antifungal
		bacillaene (100%)	antibacterial
		fengycin (86%)	antibacterial
		bacilysin (100%)	antibacterial/antifungal
		bacillibactin (100%)	antibacterial
		lanthipeptide class II	antibacterial
AFE 5	18	sessilin ^2^ (100%)	antifungal
AFE 8	14	viscosine (100%)	antifungal
		tolaasin (60%)	antibacterial/antifungal
AFE 9	5	-	
AFE 16	5	-	
AFE 21B	21	difficidin (100%)	antibacterial
		macrolactin (100%)	antibacterial/antifungal
		bacillaene (100%)	antibacterial
		fengycin (80%)	antibacterial
		bacilysin (100%)	antibacterial/antifungal
		bacillibactin (100%)	antibacterial
		surfactin (52%)	antibacterial
AFE 22	7		

^1^ Only clusters with a similarity > 50% are reported. In parentheses are similarity scores with the reference cluster. ^2^ The BGC for difficidin (AFE4A: NODE 3, NODE 2, and NODE 14) and sessilin (AFE 5: NODE 40 and NODE 26) were fragmented over different contigs and completeness was assessed after manual curation of the antiSMASH analysis.

**Table 4 pathogens-12-00013-t004:** **Siderophore production in the AFE strains.** Data converted to qualitative scale [−, no yellow halo; +, weak; ++ strong; +++ very strong].

Strain	Level of Siderophore Production
AFE 1	++
AFE 3	++
AFE 4A	−
AFE 5	+++
AFE 8	+++
AFE 9	++
AFE 11	−
AFE 14	++
AFE 16	+
AFE 21B	−
AFE 22	+++

## Data Availability

The WPB genome is available through the NCBI GenBank database under accession GCA_900102765.1. Full antiSMASH results along with AFE strain genomes can be found here: https://doi.org/10.6084/m9.figshare.21164395.v1.

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
