# Peer review of "Potential Biocontrol Activities of Populus Endophytes against Several Plant Pathogens Using Different Inhibitory Mechanisms"

_pathogens, 2022, doi:10.3390/pathogens12010013_

Round 1

Reviewer 1 Report

The title of the paper is very general for a study carried out with only one strain from each of the four fungal species studied, the results obtained may only be associated with the strain used from each species and cannot be generalized. These strains may not be the most representative of the species under study. This study should include a larger number of strains of each of the species studied from different sources and origins, including international reference strains of each of the species studied. Further, authors do not indicate in the M&M section the number of technical and biological replicates carried out in the trials, nor do they include a section on statistical analysis of the data.

In the M&M section in some cases only a citation is provided while in others where a citation is also included but the methodology is described in detail, which is not understood.  The method of endophyte isolation is also unclear. The development of a direct method for the isolation of antifungal endophytes is indicated in the abstract of the manuscript but is not well described in M&M, nor are results shown, nor comparison with standard methods. Further, the method used does not sterilise the surface of the branches, it is rather a disinfection so it is not certain that all isolates are endophytes. The culture medium is not described either, it is only said that it is a rich medium, it is all very vague.   Information must be provided so that the tests can be reproduced by other authors.

Table 1 and 5 are unnecessary, their results can be included in the text.

Author Response

1) The title of the paper is very general for a study carried out with

only one strain from each of the four fungal species studied, the

results obtained may only be associated with the strain used from

each species and cannot be generalized. These strains may not

be the most representative of the species under study. This study

should include a larger number of strains of each of the species

studied from different sources and origins, including international

reference strains of each of the species studied.

Response: The pathogen strains were selected because they are highly virulent and of concern to Pacific Northwest agriculture. The pathogen strain names have been added and we clarified that we are not claiming broad-spectrum activity for all members of the pathogen species. This study focuses on the potential for endophytes to be used in bio control rather than putting forward specific strains for commercialization.

2) Further, authors

do not indicate in the M&M section the number of technical and

biological replicates carried out in the trials, nor do they include a

section on statistical analysis of the data.

Response: We added more information on replicates numbers. Much of the data presented, however, are not of the type requiring an additional section for statistical analysis. It was primarily genomic data and presence or absence of inhibition. 

3) In the M&M section in some cases only a citation is provided while

in others where a citation is also included but the methodology is

described in detail, which is not understood.

Response: Cited methods were performed as described.  These protocols can be found in open-access articles.  When the method was performed with some modification to how it was previously described, we provided the details of the modification.

4) The method of

endophyte isolation is also unclear. The development of a direct

method for the isolation of antifungal endophytes is indicated in the

abstract of the manuscript but is not well described in M&M, nor

are results shown, nor comparison with standard methods. Further,

the method used does not sterilise the surface of the branches, it

is rather a disinfection so it is not certain that all isolates are

endophytes. The culture medium is not described either, it is only

said that it is a rich medium, it is all very vague. Information must

be provided so that the tests can be reproduced by other authors.

Response: The method used for sterilizing the surface of the branches to remove epiphytic (surface-borne) bacteria is routine for isolating endophytic (interior) bacteria.  It is called within literature, surface sterilization, even though this may be more accurately described as disinfecting. However, the harsher the treatment, the less endophytes will survive.  A true sterilization of the surface is not possible in this kind of study where live endophytes are desired. 

5)Table 1 and 5 are unnecessary, their results can be included in the

text.

Response: We merged the information into one of the other tables

Reviewer 2 Report

This manuscript described a direct screening method for isolation of anti-fungal endophytes from wild poplar extracts. Authors found that eleven isolates can inhibit two plant pathogens and were chosen for further characterization by genomic analysis of genes known to be involved in antimicrobial activities. The newly isolated Bacillus strains had gene clusters for production of bacillomycin, fengicyn, and bacillibactin, while the gene cluster for the synthesis of sessilin, viscosin and tolaasin where found in the Pseudomonas strains. The new isolates lacked the gene cluster for occidiofungin implying they employ different modes of action. In total, these results are interesting and very valuable for searching endophyte strains with potential for biocontrol applications against pathogens with high effectiveness. However, the written has a big problem, and lots of modification are needed before publishing. I marked most problems I found on the attached PDF, please download it for your revision.

Following are some minor comments:

1.A name Burkholderia vietnamiensis should be given for strain WPB in the introduction!

2.Information is lost in Table 1.

3.You can combine Table 2 and 3 into one.

4.”Using fluorescence tagging, the strains were tested for the ability to colonize crop plants. All of the strains were able to colonize multiple plant species, including kale, rice, soy, and wheat, to varying degrees”. Where are these results to be shown?

5. this title “2.7. figures” should be deleted

6. “In previous studies of endophytes that had been isolated using other screening methods, the endophytes capable of the strongest pathogen inhibition were of the genus Burkholderia [14].” For all pathogens or for all plant species? It sounds a uncorrected description, please specify this.

7.”Though strains AFE 3, 14, and 22 were originally isolated with apparent inhibitory activity against R. solani, when fully purified, they were no longer active against this pathogen. It is possible that they required a microbial partner or phytochemicals mimicking host-like conditions that were lost during the strain purification phase. For instance, the expression of T6SS, which can be used to deliver antifungal effectors, is silent unless host-like conditions are mimicked.” Here I think it is a very interesting phenomenon and worth to more discussion. For example, I think that all purified individual strains which you streak-purified from the central to the inhibition, (in particular, for those different species, varieties, genotypes), may be good candidates for a consortium of strains with complementary antifungal activities. These stains are worth to be tested possible synergies to resistance of fungal pathogen.

8.It is better to add a conclusion in the last paragraph.

9.In section Methods, the methods authors used to isolate bacteria is very valuable for searching potential antifungal pathogens as individual, or as a consortium, or acting requirement of phytochemicals. The authors mentioned that the mixed population bacterial growth central to the inhibition was then streak-purified. Any photos are better to help readers to understand.

Author Response

Reviewer #2

However, the written has a big problem,

and lots of modification are needed before publishing. I marked

most problems I found on the attached PDF, please download it for

your revision.

Following are some minor comments:

1.A name Burkholderia vietnamiensis should be given for strain

WPB in the introduction!

Response: WPB affiliation to the B. vietnamiensis species has been added in the Introduction

2.Information is lost in Table 1.

Response: We have removed the empty column in table 1, no information was lost.

3.You can combine Table 2 and 3 into one.

Response: Table 2 and 3 are now merged. See Table 2

4.”Using fluorescence tagging, the strains were tested for the

ability to colonize crop plants. All of the strains were able to

colonize multiple plant species, including kale, rice, soy, and

wheat, to varying degrees”. Where are these results to be shown?

Response: We added the information as a supplementary table.  Though it was a lot of work for the undergraduate student to get the data, it was qualitative data, simply whether or not and to what relative degree, the strain was seen in the plants

  1. this title “2.7. figures” should be deleted

Response: We were following the provided template; however, we deleted it upon this request

  1. “In previous studies of endophytes that had been isolated using

other screening methods, the endophytes capable of the strongest

pathogen inhibition were of the genus Burkholderia [14].” For all

pathogens or for all plant species? It sounds a uncorrected

description, please specify this.

Response: We changed the sentence into: “In a previous study, the endophytes showing the strongest antifungal activity were of the genus Burkholderia

7.”Though strains AFE 3, 14, and 22 were originally isolated with

apparent inhibitory activity against R. solani, when fully purified,

they were no longer active against this pathogen. It is possible that

they required a microbial partner or phytochemicals mimicking

host-like conditions that were lost during the strain purification

phase. For instance, the expression of T6SS, which can be used

to deliver antifungal effectors, is silent unless host-like conditions

are mimicked.” Here I think it is a very interesting phenomenon

and worth to more discussion. For example, I think that all purified

individual strains which you streak-purified from the central to the

inhibition, (in particular, for those different species, varieties,

genotypes), may be good candidates for a consortium of strains

with complementary antifungal activities. These stains are worth to

be tested possible synergies to resistance of fungal pathogen.

Response: Indeed, given the high diversity of BGCs involved in the synthesis of antifungal compounds, the AFE strains characterized in this study are good candidates for the development of a microbial consortium with enhanced antifungal activity against a broader spectrum of plant pathogens. However, the AFE strains WPB, AFE21B, AFE4A and AFE8 also carried gene clusters for the synthesis of antimicrobial compounds. Therefore, preliminary studies are necessary to understand possible antagonistic effects between members of the microbial consortium.  

8.It is better to add a conclusion in the last paragraph.

Response: We have added a conclusion.

9.In section Methods, the methods authors used to isolate bacteria

is very valuable for searching potential antifungal pathogens as

individual, or as a consortium, or acting requirement of

phytochemicals. The authors mentioned that the mixed population

bacterial growth central to the inhibition was then streak-purified.

Any photos are better to help readers to understand.

Response: The streak-purification method is standard for any microbiologist.  Photos of a streaked plate would be unnecessary, especially for this journal based on microbiology

Edits from Reviewer #2 edited manuscript:

Italicized the genera in the Keywords

Added another reference for a review on biocontrol

Added more information in the legend for Table 1

Figure 1 already has the labels at the top for WPB wild-type and mutant.  Added the diameter of the petri dish to the Materials and Methods

Fixed the “missing information” in Table 2 by deleting the empty column

In the tables, species names are now italicized but not the abbreviation “sp.”

The colonization data were added as a Supplementary table

Added the petri dish size to the Materials and Methods

More description on the “4 drops” in the Chemical extraction section was added

Figure A1 indicates auxin production by the endophyte strains.  We do not discuss their relative activities and so did not include a statistical analysis to compare them

Reference section was revised

Reviewer 3 Report

This is an interesting manuscript, which demonstrates antifungal effects of the bacterial endophytes obtained from Populus. However, there are some deficiencies in the MS which needs to be revised. For example, IAA and siderophore production capabilities of the strains are investigated, but are not mentioned in the abstract, which should be added. Performing microscopic analysis will be helpful for determining modes of action of the endophytic strains against each fungal pathogen tested. In addition, it is necessary to inoculate each endophytic strain on the main host plant of each fungal pathogen to investigate the possibility of producing any disease symptom on the plant tissues and also plant protection in vivo on the intact plants.  

Author Response

For example, IAA and siderophore production capabilities of the strains are investigated, but are not mentioned in the abstract, which should be added. Performing microscopic analysis will be helpful for determining modes of action of the endophytic strains against each fungal pathogen tested. In addition, it is necessary to inoculate each endophytic strain on the main host plant of each fungal pathogen to investigate the possibility of producing any disease symptom on the plant tissues and also plant protection in vivo on the intact plants. 

Response: The additional assays were added to the Abstract.

Performing microscopic analysis will be helpful for determining modes of action of the endophytic strains against each fungal pathogen tested.

Response: We agree that it could be interesting to visually investigate the modes of action. Now that we have completed the genomic analysis for potential known mechanisms, further investigations can include this method for beginning to determine the unknown mechanisms.

In addition, it is necessary to inoculate each endophytic strain on the main host plant of each fungal pathogen to investigate the possibility of producing any disease symptom on the plant tissues and also plant protection in vivo on the intact plants.

Response: The next phase of this research towards the development of potential biocontrol strains for use in agriculture will be to do as the Reviewer suggests. 

Reviewer 4 Report

So far studies on plant benificial bacteria are mainly associated with agricultural plants and some model plants. Much less is performed with woody plants. In this article Doty et al isolated some endophytes from populus and investigated their boncontol activities revealing different inhibitory mechanism. The paper is very interesting to me and I think it is suitable for publication on Pathoghens.

Just some minors: 

1. : the title is obviously not complete. Populus endophytes should be included in the title. I also recommend to replace "agrcicultural" with "plant". 

L80. not clear.  How can endophytes have an advantage over some fungal endophytes?

Author Response

  1. the title is obviously not complete. Populus endophytes should be included in the title. I also recommend to replace "agrcicultural" with "plant". 

Response: Yes, the title has been in flux through review.  In the previous round of revisions, the word Populus was removed from the title.  We now propose a new title: “Potential biocontrol activities of Populus endophytes against several plant pathogens using different inhibitory mechanisms”

L80. not clear.  How can endophytes have an advantage over some fungal endophytes?

Response: We added the word “bacterial” to describe the type of endophytes to which we were referring, and added a brief explanation that fungal endophyte can be highly localized.

Reviewer 5 Report

The manuscript is interesting and well-written. Results are well analyzed and discussed. However, two methodological mistakes are still widespread but that need to be clarified.

Here are my comments:

1.      Page 9, lines 307-309, page 14, line 574, IAA production: Salkowski’s method is used because it is easy under common laboratory conditions. While it indicates the presence of the auxin produced by the bacteria, verification and quantification of IAA must be done by a specific HPLC method; otherwise, it shows only IAA-like molecules production. Thus, this needs to be clarified in Methods and Results.

2.      Page 9, line 308, page 14, line 574, The use of NBRIP medium as a sole source to determine phosphate solubilization is a methodological mistake. In the literature of in vitro screening, it is common to use triphosphate calcium as the substrate, although it has very little to do with reality because strains capable of degrading it are not necessarily PSBs that promote plant growth. To establish the capacity of phosphorus solubilization by a strain, there is need to test with a harder-to-dissolve compound, such as Fe-P or Al-P (see Biol. Fertil. Soils 2013, 49:465-479). This limitation needs to be highlighted in Methods and Results. Also, the results of tricalcium phosphate solubilization are not presented (page 9, lines 310-311)

3.      The ability of the endophyte strains to colonize crop plants (page 14, lines 587) is poorly described. Information about the experimental design is missing: How many plants (replicates) were used, what were the growth conditions, and how were the strains inoculated? Was the assay repeated to ensure reproducibility? In Table A3, there is a need to define to what level of colonization corresponds the number of asterisks (number of plants? area of colonization?)

Additionally, the hypothesis must be added before the aims of the work.

Author Response

  1. Page 9, lines 307-309, page 14, line 574, IAA production: Salkowski’s method is used because it is easy under common laboratory conditions. While it indicates the presence of the auxin produced by the bacteria, verification and quantification of IAA must be done by a specific HPLC method; otherwise, it shows only IAA-like molecules production. Thus, this needs to be clarified in Methods and Results.

Response: We replaced the word “IAA” with “auxins” in the Results and Methods sections

  1. Page 9, line 308, page 14, line 574, The use of NBRIP medium as a sole source to determine phosphate solubilization is a methodological mistake. In the literature of in vitro screening, it is common to use triphosphate calcium as the substrate, although it has very little to do with reality because strains capable of degrading it are not necessarily PSBs that promote plant growth. To establish the capacity of phosphorus solubilization by a strain, there is need to test with a harder-to-dissolve compound, such as Fe-P or Al-P (see Biol. Fertil. Soils 2013, 49:465-479). This limitation needs to be highlighted in Methods and Results. Also, the results of tricalcium phosphate solubilization are not presented (page 9, lines 310-311)

Response: We agree that if the focus of the paper were on such symbiotic traits, we would have assayed the other forms of phosphate as we had done in a recent manuscript (Varga 2020).  For this biocontrol paper, we did only rapid tests for other potentially symbiotic traits.  We modified the text to only TCP solubilization rather than phosphate solubilization in general and indicated that more forms of phosphate should be tested for a complete analysis as described in Bashan 2013.

  1. The ability of the endophyte strains to colonize crop plants (page 14, lines 587) is poorly described. Information about the experimental design is missing: How many plants (replicates) were used, what were the growth conditions, and how were the strains inoculated? Was the assay repeated to ensure reproducibility? In Table A3, there is a need to define to what level of colonization corresponds the number of asterisks (number of plants? area of colonization?)

Response: Since the colonization study was a minor component of this manuscript, we removed it to maintain the focus on mechanisms of potential biocontrol activities.  We removed Table A3 and reference to the experiment in the Discussion.

  1. Additionally, the hypothesis must be added before the aims of the work.

Response: A statement of the hypothesis was added at the beginning of the last paragraph in the introduction (line 95-96).

Round 2

Reviewer 1 Report

Although the authors have clarified some but not all aspects, the title is still too general for the results shown: they only use one strain of each pathogen and therefore the results cannot be extrapolated to the species level, nor can they be concluded at the species level.No further work has been carried out and no additional strains have been tested to rule out that the results obtained are associated only with the strain tested.

Author Response

We have done our best to remove species-level generalities. The title was modified, and we added wording describing this study as in vitro throughout the manuscript.  We added the pathogen strain names to the Abstract, figures, and Discussion. Carrying out exhaustive bio-assays with additional pathogen strains is not feasible now that the small grant which funded this research has run out.  We begin the Discussion emphasizing that the pathogen strains that we tested were the strains of concern in our region.

Reviewer 3 Report

It is necessary to inoculate each endophytic strain on the main host plant of each fungal pathogen to investigate the possibility of producing any disease symptom on the plant tissues and also plant protection capability of the endophyte in vivo on the intact plants. 

Author Response

In regard to the request to test the endophyte strains on the main host of each pathogen, these are broad host range pathogens.  We included wheat, which is a host to all of the pathogen strains we tested. We added the words “in vitro” throughout the text when we referred to the inhibition to make it clearer that this work represents the first phase of the research and does not include in vivo biocontrol tests in agricultural settings.  In this manuscript, we describe the initial isolation of the endophyte strains, in vitro bio-assays against several pathogenic strains of concern in our region, full genomic sequencing and analysis for potential mechanisms, mass spectrometry, mutant construction and analysis, and microbiological assays to test for other symbiotic traits.  Experiments in planta are beyond the scope of the small grant we had from our university, and furthermore, the greenhouses at our university are not USDA-approved for such pathogen studies.  Such testing must be done at an agricultural university.  Although one of our authors is from such an ag university (Washington State University), she is retired and had served only in an advisory role for our study.